# In Vivo Study on the Salivary Kinetics of Two Probiotic Strains Delivered via Chewing Gum

**DOI:** 10.3390/microorganisms13040721

**Published:** 2025-03-24

**Authors:** Silvia Cirio, Claudia Salerno, Simone Domenico Guglielmetti, Valerio Mezzasalma, Andrea Sarrica, Natalja Kirika, Guglielmo Campus, Maria Grazia Cagetti

**Affiliations:** 1Department of Biomedical, Surgical and Dental Sciences, University of Milan, Via Beldiletto 1, 20142 Milan, Italy; claudia.salerno@students.unibe.ch (C.S.); maria.cagetti@unimi.it (M.G.C.); 2Department of Restorative, Preventive and Pediatric Dentistry, University of Bern, Freiburgstrasse 7, 3012 Bern, Switzerland; 3Graduate School for Health Sciences, University of Bern, 3012 Bern, Switzerland; 4Department of Biotechnology and Biosciences, University of Milano-Bicocca, Piazza Della Scienza 2, 20126 Milan, Italy; simone.guglielmetti@unimib.it (S.D.G.); valerio.mezzasalma@fem2ambiente.com (V.M.); 5Department of Research and Scientific Affairs, Perfetti Van Melle S.p.A., Lainate, 20045 Milan, Italy; andrea.sarrica@perfettivanmelle.com (A.S.); natalja.kirika@perfettivanmelle.com (N.K.); 6Department of Cariology, Institute of Odontology, Sahlgrenska Academin, University of Gothenburg, 40530 Gothenburg, Sweden; guglielmo.giuseppe.campus@gu.se; 7Department of Oral and Maxillofacial Sciences, Sapienza University of Rome, 00185 Roma, Italy; 8Department of Cariology, Saveetha, Dental College and Hospitals, SIMATS, Chennai 600077, India

**Keywords:** probiotic, sugar-free chewing gum, saliva, *Heyndrickxia coagulans*, *Lacticaseibacillus rhamnosus*

## Abstract

Probiotics are increasingly used to promote oral health, with *Lacticaseibacillus rhamnosus* demonstrating proven effectiveness. Additionally, *Heyndrickxia coagulans* shows promising potential in this field. Chewing gum has recently been proposed as an innovative delivery method for probiotics. This study aimed to evaluate the kinetics in saliva of *Heyndrickxia coagulans SNZ1969*^®^ and *Lacticaseibacillus rhamnosus GG* in microencapsulated and non-microencapsulated forms (LGG^®^) following their administration via sugar-free chewing gums. A randomized cross-over trial was conducted involving 10 volunteers. Participants chewed gums containing one of the probiotic strains for 10 min. Saliva samples were collected at baseline (T_0_) and six subsequent time points over 2 h (T_1_–T_6_). Colony-forming units (CFUs) were identified and quantified. The Tukey’s range test was applied to make pairwise comparisons between different probiotics at every time point, between different time points of the same probiotic, and between the area under the curve describing the kinetics of different probiotics in saliva. At T_1_, all probiotics exhibited peak counts, followed by a gradual decline until T_6_. *H. coagulans* SNZ1969^®^ achieved the highest counts at T_1_, T_2_, and T_3_ (mean log_10_ CFU/mL: 6.1 ± 0.5; 5.8 ± 0.5; 5.6 ± 0.5, respectively), while the non-microencapsulated form of LGG^®^ peaked at T_4_, T_5_, and T_6_ (mean log_10_ CFU/mL: 4.0 ± 0.7; 3.8 ± 0.9; 3.3 ± 1.3, respectively). The participants reported no adverse effects. Probiotics were detectable in saliva up to 2 h post-administration via chewing gum, indicating its suitability as a delivery vehicle. However, significant variability was observed among participants.

## 1. Introduction

The oral cavity is intimately connected to other systems and organs, as evidenced by associations between oral health and conditions such as cardiovascular disease, diabetes, lung disease, and pregnancy [1]. These associations can be ascribed to several factors, including alterations in the community structure of the oral microbiota.

Probiotics have become widely used over the last two decades. They are defined as “live microorganisms that, when administered in adequate amounts, confer a health benefit to the host” [2]. Initially recognized for their gastrointestinal benefits, probiotics have been extensively studied for their ability to prevent and treat gastrointestinal diseases, enhance digestive well-being, and combat pathogens [3,4,5,6,7,8]. These benefits contributed to a growing awareness of their positive effects and a subsequent increase in their use [9]. Consequently, the global probiotics market is projected to grow from USD 71.2 billion in 2024 to USD 105.7 billion in 2029, at an annual growth rate of 8.2% expected over this period [10].

Beyond gastrointestinal health, probiotics have demonstrated potential benefits for oral health, including caries, halitosis, and periodontitis, in healthy individuals and those with systemic diseases [11,12,13]. Conditions such as dental caries and periodontal disease have been associated with an imbalance in the oral microbiota. This association is explained by the “ecological plaque hypothesis” [14] and probiotic administration has shown promise in addressing this imbalance. Some of the most studied and promising strains include *Lactobacillus rhamnosus*, *Lactobacillus reuteri*, *Lactobacillus acidophilus*, *Lactobacillus salivarius*, *Lactobacillus casei* and *paracasei*, *Bifidobacterium lactis*, and various others [15]. As evident from the above, all strains have demonstrated beneficial effects on the oral cavity. *Lacticaseibacillus rhamnosus* GG is one of the most extensively investigated, and it has shown its ability to counteract the most prevalent diseases of the gums and teeth, dental caries, and periodontal disease [16,17,18,19]. In addition, it has shown an inhibitory effect on halitosis-causing bacteria, such as *Porphyromonas gingivalis*, *Tannerella forsythia*, and *Prevotella intermedia* [20] and a strong anticandidal activity [21]. All these properties make it a remarkably versatile and effective microorganism for promoting oral health and preventing common oral diseases.

Spore-forming probiotics are experiencing an increase in popularity, attributable to their capacity to enhance survival and stability. In the field of functional food research related to human health, there is an increasing focus on *Bacillus* spp. due to their remarkable tolerance and survivability in the harsh conditions of the gastrointestinal tract. Furthermore, their superior stability during food and pharmaceutical processing and storage renders them ideal candidates for health-promoting formulations [22]. In contrast, vegetative probiotic species are more sensitive to these processes and often require refrigeration to maintain their potency [23]. Microencapsulation is proposed as an effective method for protecting probiotics, enhancing their viability during industrial processing, and extending their stability during storage and digestion [24].

*Heyndrickxia coagulans* (formerly *Bacillus coagulans*) has demonstrated antimicrobial, antioxidant, and immunomodulatory properties [25], and it has recently received considerable attention in dentistry [26,27]. Recent studies highlight its effectiveness in controlling dental caries by reducing *Streptococcus mutans* and *Lactobacillus* spp. counts in plaque and saliva [23,26]. Additionally, they have been shown to lower gingival index scores, reduce bleeding on probing, and combat gingival inflammation [28]. *H. coagulans* is listed by the European Food Safety Authority (EFSA) under the Qualified Presumption of Safety status for recommended biological agents. Its use has been approved due to the absence of acquired antimicrobial resistance genes to clinically relevant antibiotics and the lack of toxigenic activity [29], unlike other *Bacillus* spp. [30].

Sugar-free chewing gum positively affects oral health by stimulating saliva flow and promoting natural oral clearance mechanisms [31,32,33,34]. Moreover, it can serve as a delivery system for ingredients such as xylitol or fluoride, active in reducing dental plaque and the concentration of cariogenic bacteria, such as *Streptococci mutans*, and in remineralizing enamel, respectively [35,36]. The EFSA (European Food Safety Authority) has acknowledged the potential benefits of chewing sugar-free gum for oral health, including maintaining tooth mineralization, neutralizing plaque acids, and the reduction in oral dryness. To reap these benefits, the EFSA has recommended that individuals chew 2–3 g of sugar-free gum for at least 20 min, at least three times per day following mealtimes [37,38,39]. Sugar-free chewing gums are composed of a gum base, with the sugar being entirely replaced by alternative bulk sweeteners consisting of one or a combination of several polyols, such as sorbitol, mannitol, isomalt, maltitol, maltitol syrup, lactitol, xylitol, and erythritol. High-intensity sweeteners, including acesulfame K, aspartame, cyclamic acid and its sodium and calcium salts, saccharin and its sodium, potassium, and calcium salts, sucralose, thaumatin, neohesperidine dihydrochalcone, and the aspartame–acesulfame salt, are commonly used alone or in combination with other food additives and flavors [40]. Moreover, chewing gum is experiencing a surge in popularity as an oral drug delivery system, owing to its ease of use and palatability. Its applications extend to the domains of pain management, smoking cessation, and anti-emetic therapies. [41,42]. Chewing gum is particularly suitable for school children and adolescents, given the high compliance rates observed in these age groups [18].

The administration of probiotics through sugar-free chewing gum has demonstrated encouraging results in enhancing oral health by substantially reducing plaque accumulation, gingival scores, *S. mutans* counts, and bleeding on probing. Furthermore, a reduction in inflammatory mediator levels in gingival crevicular fluid, a major indicator of periodontal disease, has been observed. In addition to these benefits, probiotics have also been found to assist in the alleviation of halitosis (bad breath) [43,44,45,46,47]. The benefits mentioned above suggest that delivering probiotics via chewing gum could be an effective adjunct in managing oral conditions. Although studies have demonstrated the clinical efficacy of sugar-free probiotic chewing gum, it is imperative to ascertain the amount of the administered probiotic that remains in the oral cavity, as this directly influences its ability to colonize oral surfaces and ensure a lasting effect [48,49]. While the release and concentration of various active agents delivered via chewing gum have been investigated in both in vitro and in vivo studies [36,50], no research has evaluated the release kinetics of probiotics from chewing gum to date. Therefore, an in vivo microbiological study was designed to analyze the kinetics of probiotics in saliva in 2 h follow-up, following their administration through sugar-free chewing gum. The hypothesis to be tested is that the probiotic administered through chewing gum remains in saliva long enough for it to adhere to oral surfaces [51]. This study constitutes the initial phase of a more comprehensive research initiative. The subsequent stage of the project will entail an evaluation of the capacity of probiotics administered via sugar-free chewing gum to colonize oral surfaces and modify the microbial diversity of the oral microbiome.

## 2. Materials and Methods

### 2.1. Design of the Study

This randomized, cross-over microbiological study was designed and conducted at the Department of Biomedical, Surgical and Dental Sciences, University of Milan, Milan, Italy, following the principles of the Declaration of Helsinki. The Ethical Committee of the University of Milan approved the study (13 February 2024, no. 24/24). Recruitment of participants, intervention, and microbiological measurements were carried out between March and May 2024. This study represents the initial phase of a larger research project designed to evaluate the ability of probiotics delivered via chewing gum to colonize the oral cavity and dental plaque.

### 2.2. Sample Selection

The study was conducted on healthy adult volunteers selected from the staff of the Perfetti Van Melle S.p.A. (Lainate, Milan, Italy).

No prior studies with the same objective were identified, so a sample size calculation could not be performed. Therefore, conducting an in vivo microbiological study with 10 participants was decided. The inclusion criteria were adult subjects aged 18 to 64 years, at least 24 natural teeth (excluding third molars), gingival index and plaque index scores ≤ 2, and a stimulated salivary flow rate between 1.5 and 2.0 mL/min. Exclusion criteria included the presence of systemic diseases, pregnancy or lactation, history of drug abuse, smoking habits, use of fixed orthodontic appliances, and allergies to any ingredients in the chewing gums used. An email explaining the purpose of the study and inviting participation was sent to all staff at the PVM site in Lainate. Thirteen individuals who consented to participate were interviewed to assess their eligibility based on the inclusion and exclusion criteria. They were then examined by a calibrated dentist (SC) to obtain their gingival index scores [52], gingival index [53], and stimulated salivary flow rate. Ten eligible subjects were identified and enrolled. All study participants gave their written consent to participate.

### 2.3. Chewing Gums Production

All chewing gums used in the study were produced and supplied by PMV (Appendix A). The sugar-free chewing gums (weight 2.1 g) were formulated with gum base (Gum Base Co., Lainate, Milan, Italy), food-grade polyols, excluding xylitol (proprietary blend; manufactured by Roquette Frères S.A., Cassano Spinola, Alessandria, Italy and Cargill S.r.l., Milan, Italy), food-grade intensive sweeteners (Ajinomoto Co., Inc., Tokyo, Japan), flavors (Mondarom Selegroven AG, Bironico, Switzerland), and incorporated specific probiotic strains under investigation. These included *Lacticaseibacillus rhamnosus* GG provided either in its non-microencapsulated form (LGG^®^, DSMZ code: DSM 33156, supplied by Chr. Hansen, Boege Alle 10-12, 2970 Hoersholm, Denmark) or as microencapsulated cell (Encaptimus^TM^, ATCC 53103, provided by AnaBio Technologies Ltd., 11 Herbert Street, Dublin, D02 RW27, Ireland; containing Maltodextrin, *Lactobacillus rhamnosus*, Coconut Oil, Pea Protein Isolate, Polysorbate 20) [54], and *Heyndrickxia coagulans* SNZ1969 (*Heyndrickxia coagulans* SNZ1969^®^, provided by Sanzyme Biologics Ltd., Sattva Signature Tower, H. No. 8-2-472/1/A/B/SF-3, Road No. 1, Banjara Hills, Hyderabad, 500034 Telangana, India), which was added in its spore form. Therefore, three different chewing gums were produced.

The production process of the chewing gum is summarized in the Appendix A. Initially, the gum base is melted at 50 °C and combined with polyols, artificial intense sweeteners, and flavorings sequentially, achieving a homogeneous mixture. The freeze-dried probiotic biomass is incorporated as a final component below 50 °C to preserve its activity. After mixing, in the rolling and scoring system, a mass of gum is extruded into a thick slab, which is then worked into a thinner and thinner foil by a series of rollers. Finally, the foil is shaped into single pieces by one or more cutting rollers. These pieces undergo cooling in a conditioning room before being panned, and afterward, individual pieces are prepared for packaging [55].

At the end of the production process, the amount of Colony-Forming Units (CFUs) contained in the chewing gum was assessed for each probiotic strain according to the following protocol. The gum pieces in the bag were manually crushed and then processed in a Stomacher^®^ (Stomacher^®^ 3500 peristaltic homogenizer, Seward, West Sussex, UK) for 2 min. The resulting homogenized chewing gum mass was further serially diluted 1:10 in Maximum Recovery Diluent (MRD) buffer (GranuCult^®^ prime Peptone salt solution—Maximum Recovery Diluent, Merck KGaA, Darmstadt, Germany). Aliquots of 100 μL were plated onto selective media: GYEA-agar (Glycerol 5 g/L, Yeast extract 2 g/L, K_2_HPO_4_ 1 g/L, BromoCresol Green 10 mL/l from a stock of 5 g/L, Agar 15 g/L pH 5.5, Merck KGaA, Darmstadt, Germany) for *H. coagulans* SNZ1969^®^ and RVB-MRS-agar (Lactobacilli MRS *w*/*o* Dextrose 3515 g/L, Condalab, Madrid, Spain; Rhamnose 20 g/L, Bromocresol green 10 mL/l from a stock of 5 g/L, Agar 15 g/L pH 5.5 after autoclave, Cysteine-HCl 0.05%, Vancomycin 50 mg/L, Merck KGaA, Darmstadt, Germany) for *L. rhamnosus* GG). For *H. coagulans* SNZ1969^®^, part of the samples underwent viable count after pasteurization (incubation in a water bath at 90° C for 10 min) to quantify bacterial spores. The plates were incubated at 37° C for 72 h for the evaluation of *L. rhamnosus* GG and at 55° C for 72 h in anaerobic conditions, established by incubating the plates in AnaeroJar Oxoid 2.5 L jars (Thermo Fisher Scientific™, Waltham, MA, USA) containing Anaerocult™ A (Merck KGaA, Darmstadt, Germany), for the assessment of *H. coagulans* SNZ1969^®^ CFUs. The procedures described were repeated for each sample in triplicate. CFUs were identified by morphology and color and finally counted. The viable bacterial cell counts were expressed as colony-forming units/gram (CFU/g).

At the end of the production process, the mean counts of probiotics in one pellet of chewing gum were as follows:-6 × 10^8^ CFU of *Lacticaseibacillus rhamnosus* LGG^®^ (non-microencapsulated form);-2 × 10^8^ CFU of *Lacticaseibacillus rhamnosus* GG (microencapsulated);-5 × 10^8^ CFU of *Heyndrickxia coagulans* SNZ1969^®^.

### 2.4. Use of Chewing Gum

Participants were instructed to chew a pellet of gum containing one of the probiotic strains. Following a washout period of one week, they were asked to chew a second pellet containing a different strain. After another one-week washout period, they chewed the third and final pellet. At each visit, each subject was administered only one formulation per session in a randomized cross-over design to ensure that the efficacy of each formulation was assessed independently. Each subject drew one of three chewing gums at the first appointment using a lottery system. They drew one of the two remaining gums at the second appointment, and the last remaining gum was administered at the third appointment. Administration occurred in the morning, at least 2 h after breakfast and oral hygiene routines.

Both participants and investigators were blinded to the specific probiotic strain in each gum. Volunteers were instructed to chew the gum for 10 min and to refrain from eating or drinking anything for the subsequent 2 h.

Saliva samples of at least 0.5 mL were collected from the floor of the mouth using sterile disposables at the following time points: before chewing gum use (T_0_), and at 1, 5, 10, and 20 min, as well as 1 and 2 h after the procedure began (T_1_–T_6_). Samples were stored at 4 °C, transported to the laboratory, and processed within 2 h.

An investigator (SC) conducted brief telephone interviews with the participants on the evening of the intervention day and one week later to record any side effects associated with the chewing gum administered.

The flow chart of the study design is presented in Figure 1. At the first appointment, each participant chewed one randomly selected chewing gum from the three gums tested in the study. At the second appointment, they chewed one of the two remaining gums, and at the final appointment, the last remaining gum. This process was repeated for each participant, ensuring that all three gums were tested in a randomized order across the different subjects. Each chewing gum was chewed 2 h after breakfast and the oral hygiene routine. Salivary samples were then collected over 2 h and analyzed microbiologically.

### 2.5. Microbiological Analyses

Aliquots of 300 μL of saliva were diluted in 600 μL of MDR buffer (GranuCult^®^ prime Peptone salt solution—Maximum Recovery Diluent, Merck KGaA, Darmstadt, Germany). Samples were processed as it was explained for chewing gum analysis, but without any preventive treatment to break down the microcapsules. If different morphologies were detected, three colonies per type were selected and analyzed by colony polymerase chain reaction (PCR), picking the colony into a PCR reaction with the strain-specific primers PVM-Wc-1F 5′-TTGTCTTTGGATCAGTTACAG-3′ and PVM-Wc-1R 5′-GCATAGGAATACCTTGTGCA-3′ for *H. coagulans* SNZ1969^®^ [56] and the primers GG I 5′-CAATCTGAATGAACAGTTGTC-3′ and GG II 5′-TATCTTGACCAAACTTGACG-3′ for *L. rhamnosus* GG [57]. Morphologies that revealed expected amplicons by agarose gel electrophoresis were confirmed as CFUs and included in the final count. Some of the amplicons obtained were confirmed by Sanger sequencing [58]. The amount of viable *H. coagulans* SNZ1969^®^ (without spores) was obtained by subtracting the total *H. coagulans* colony count minus the colonies of pasteurized *H. coagulans* SNZ1969^®^ (spores) for each interval.

### 2.6. Statistical Analysis

All data were transformed into a logarithmic scale to normalize the distribution. Values between 0 and 1 were rounded to 1 before the log transformation.

The ANOVA test was applied. When the variance, evaluated with Bartlet’s equal-variances test, was not equal, Welch’s *t* test was used to assess differences between probiotics. Tukey’s range test was applied to make pairwise comparisons between different probiotics at every time point, different time points of the same probiotic, and between the area under the curve of different probiotics. The area under the curve was calculated to quantify the overall exposure of the subject’s oral cavity to probiotic strains over time using the following STATA commands: mean_log_counts of the different probiotics and then the mean trapezoid_area, over the probiotic counts. Cuzick’s test with rank scores was used to describe trends within each strain during time. All data were analyzed using STATA^®^ software (v18 for Mac). Statistical significance was set at α = 0.05 for all analyses.

## 3. Results

All included subjects, 6 females and 4 males, age range 23–52 years (mean age 36.4 ± 10.0), completed the study. No adverse effects were reported by the participants or noted by the investigators during the intervention.

At T_0_, none of the probiotics tested was detected in saliva. At T_1_, the highest counts’ value was registered for all probiotic strains, followed by a constant decrease until T_6_.

*L. rhamnosus* GG in microencapsulated form showed the lowest salivary counts at each time point (*p* < 0.01) (Appendix A; Figure 2).

When comparing *H. coagulans* SNZ1969^®^ and *L. rhamnosus* LGG^®^ in non-microencapsulated form, the total viable counts of *H. coagulans* SNZ1969^®^ (i.e., spores + vegetative cells) were higher at T_1_, T_2_, and T_3_. In comparison, the viable counts of *L. rhamnosus* LGG^®^ were higher at T_4_, T_5_, and T_6_. However, no significant differences were observed between the two probiotic strains (*p* = 0.64 at T_1_; *p* = 0.52 at T_2_; *p* = 0.24 at T_3_; *p* = 0.67 at T_4_; *p* = 0.08 at T_5_; *p* = 0.02 at T_6_) (Appendix A; Figure 2).

At T_1_, T_4_, T_5_, and T_6_, the proportion of *H. coagulans* SNZ1969^®^ in vegetative form was higher than that in spore form, but a significant difference was observed at T_4_, 20 min after the start of chewing (3.3 ± 0.6 log_10_ CFU/mL vs. 2.3 ± 1.2 log_10_ CFU/mL; *p* = 0.01) (Appendix A; Figure 2).

A high variability in the salivary counts among participants was observed for all the probiotic strains tested (Figure 2; Appendix A).

All probiotic strains exhibited significant declines in salivary counts from T_1_ to T_6_ (*p* < 0.01) (Appendix A). Notably, the patterns differed between *L. rhamnosus* GG and *H. coagulans* SNZ1969^®^. For both microencapsulated and non-microencapsulated forms of *L. rhamnosus* GG, the counts’ differences between consecutive time points were not statistically significant, indicating a consistent linear decrease (Appendix A). In contrast, the total counts of *H. coagulans* SNZ1969^®^ (counts of vegetative forms + spores) and its pasteurized form (spore counts) showed a rapid decline between T_3_ and T_4_ (*p* < 0.01). The total counts of SNZ1969^®^ also exhibited a significant drop between T_5_ and T_6_ (*p* < 0.01) (Appendix A). However, no significant differences were observed at each time point for *H. coagulans* SNZ1969^®^ with spores excluded (Appendix A).

Considering the area under the curve, a significant difference was detected between *L. rhamnosus* LGG^®^ in non-microencapsulated form, *L. rhamnosus* GG in microencapsulated form, and *H. coagulans* SNZ1969^®^ (i.e., spores + vegetative cells) (*p* < 0.01). Otherwise, significant differences between *L. rhamnosus* LGG^®^ and *H. coagulans* SNZ1969^®^ (i.e., spores + vegetative cells) were not detected (*p* = 0.90) (Figure 3).

## 4. Discussion

This viable microbiological study investigates the kinetics of *L. rhamnosus* GG and *H. coagulans* SNZ1969^®^ released from sugar-free chewing gums. Despite considerable variability in salivary counts among participants, the highest bacterial amount was attained after 1 min of chewing, with levels persisting at relatively high counts for up to 10 min. In some subjects, probiotics were still detectable in saliva approximately 2 h after chewing commenced.

Although there is a paucity of research in this area, studies have been conducted on the oral health benefits of probiotics administered via chewing gum. However, these studies have not specifically examined the kinetics of probiotics in saliva, focusing on their effects on oral biofilm, such as reducing *S. mutans* counts in saliva or dental plaque [45,59].

Few studies have evaluated the kinetics of probiotics obtained with different vehicles. Lozenges were shown to expose the oral environment to probiotics for an estimated period of tens of minutes [45,60]. The utilization of a mucoadhesive lipogel comprising probiotics resulted in a substantial and consistent release for 5 to 8 h in an in vitro model [61]. In the present study, chewing gum appears to promote the presence of probiotics in saliva for a longer time than lozenges but shorter than those obtained by lipogel in vitro. As anticipated, the probiotic load diminished following the gum mastication, yet it remained quantifiable in most subjects 2 h post-ingestion. This comparatively protracted residence time in the oral environment may play a pivotal role in plaque colonization by the delivered bacteria.

Active agents administered via sugar-free chewing gum exhibited analogous kinetic trends in saliva. The investigation focused on the salivary concentration of xylitol and fluoride produced by sugar-free chewing gum [36,62], which showed a salivary peak in the first 5 min, followed by a rapid decrease in the next 5 min and a slow decline in the tens of minutes to follow. The salivary counts of the probiotics tested in this study initially peaked analogous to that of xylitol and fluoride. However, this was followed either by a gradual decrease, as seen with *L. rhamnosus* GG, or by a later but rapid decline (after 10 min), as observed with *H. coagulans* SNZ1969^®^. The decline in probiotic counts is mainly due to swallowing, which is intensified by the increased salivary flow stimulated by chewing gum. Additionally, the potential bactericidal action of salivary molecules cannot be ruled out [31,32,33,34].

The significant variability observed in the salivary counts of the probiotics among the ten study participants is a matter of concern. Despite the inclusion criterion of normal salivary flow, it can be hypothesized that such substantial inter-individual differences are not related to the efficacy of chewing gum in releasing probiotic strains. Instead, they are likely to be attributable to the individual characteristics of the participants, such as the strength and number of chewing movements per unit of time [63]. Additionally, the different stimulation of saliva among subjects caused by the fruit flavoring in the chewing gum could further explain the observed variability [64].

One aspect that has been the focus of academic inquiry is microencapsulation’s protective effect on chewing gum’s bacterial survival. The study revealed that microencapsulation significantly enhanced the survival of *Limosilactobacillus reuteri* in chewing gum over 21 days. The protective role of inulin and lecithin in this process has been postulated [65]. In a recent study, microencapsulation improved the viability of *Bifidobacterium animalis* subsp. *lactis* but not of *Levilactobacillus brevis* [61,66]. Although the encapsulated form may offer advantages in terms of microbial survival within the product, the results of the present study indicate that *L. rhamnosus GG* in its microencapsulated form produced a lower salivary count than the non-microencapsulated form, and by far the lowest among the probiotics tested. This finding suggests that the enhanced survival observed in the chewing gum product is counterbalanced by a diminished capacity to release the strain in its non-microencapsulated form. Consequently, probiotics in microencapsulated form may be advantageous for gastrointestinal effects but may not be as effective for promoting oral health.

There are certain limitations to this study. The first limitation is the limited sample size, which limits the generalizability of the results to a larger population [67,68]. Probiotic studies designed and conducted for gastrointestinal application usually had a low number of participants as they focus solely on the microbiological performance of the probiotic strain, independent of any direct effects on the host. The present research aligns with this established approach, as its primary objective was to quantify probiotic cells’ presence in saliva after chewing gum administration. Given this specific focus, the sample size is fully consistent with existing literature. Numerous published works in this domain have employed similar sample sizes, as the primary endpoint (bacterial recovery) does not require the large cohorts typically associated with clinical efficacy trials. Furthermore, participants were selected according to rigorous criteria (e.g., number of teeth, salivary flow, and oral health status) to minimize uncontrolled variability, and the cross-over design enhances the reliability of the results of this study by allowing within-subject comparisons, thereby reducing inter-individual variability [66].

In the present study, it was ensured by verifying that none of the participants had detectable counts of the tested probiotic strains in their saliva before chewing the gum (T_0_ time point). A control group using a probiotic-free chewing gum would not provide additional meaningful information in this context, as this study does not aim to evaluate clinical effects or host responses but rather the presence of the administered probiotic in saliva. The presence or absence of probiotics in post-administration samples can be directly attributed to the intervention itself, as confirmed by the absence of these strains in baseline samples.

Further research involving larger cohorts is necessary to draw definitive conclusions regarding the kinetics of probiotics released from sugar-free chewing gum, as this factor may significantly influence their functionality within the oral environment. Furthermore, the considerable variability observed in the counts of probiotics among the study participants requires further investigation, as it could substantially impact their efficacy. The present research shows that some individuals displace bacteria more quickly than others, underscoring the necessity for personalized approaches when contemplating probiotic interventions, as individual differences can influence the effectiveness of such treatments.

## 5. Conclusions

This study demonstrated that a 10 min use of sugar-free chewing gum containing probiotics led to their detectable presence in saliva. *L. rhamnosus* GG and *H. coagulans* SNZ1969^®^ exhibited comparable salivary counts, suggesting that both strains are well-suited for this mode of administration.

These findings represent the initial phase of a broader research initiative. It is well known that a single administration of probiotics is not capable of permanently modifying the composition of the oral bacterial flora [69]. Further investigations are needed to determine the ideal duration of the probiotic’s presence in the oral cavity to support colonization of the oral biofilm, thus improving its long-term efficacy. Moreover, the ability of probiotics administered via sugar-free chewing gum to influence microbial diversity following prolonged administration will be evaluated in the next part of this research.

## Figures and Tables

**Figure 1 microorganisms-13-00721-f001:**
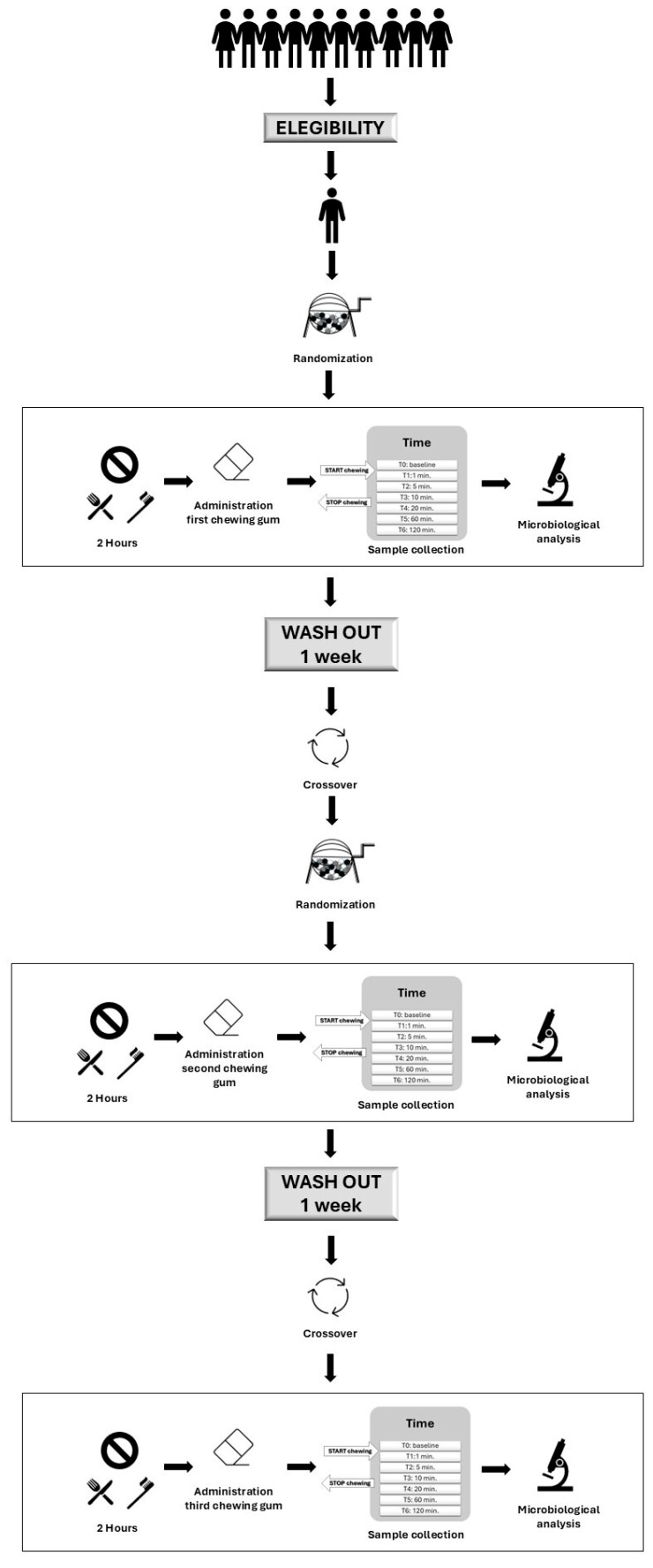
Flow chart of the study design.

**Figure 2 microorganisms-13-00721-f002:**
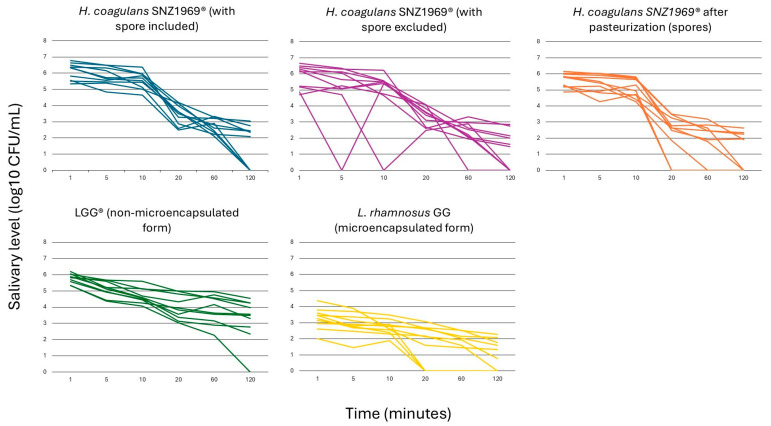
Viable count kinetics of probiotic cells in salivary samples.

**Figure 3 microorganisms-13-00721-f003:**
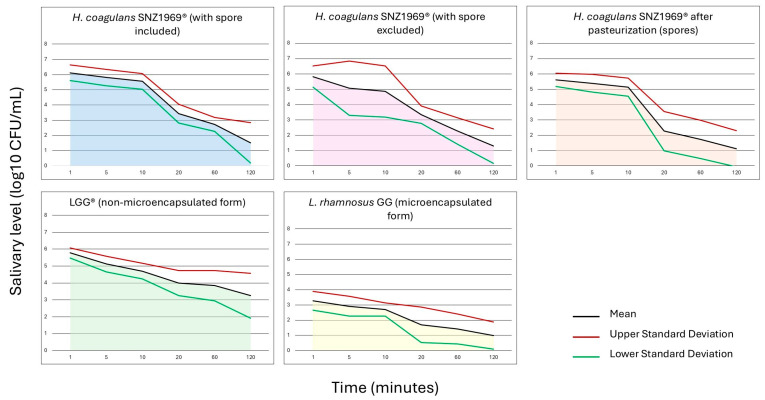
Area under the curve of probiotic cells in salivary samples.

## Data Availability

Data are available on reasonable request by contacting the corresponding author to not disclose information as we are patent pending.

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
