# Peer review of "In Vivo Study on the Salivary Kinetics of Two Probiotic Strains Delivered via Chewing Gum"

_microorganisms, 2025, doi:10.3390/microorganisms13040721_

Round 1
Reviewer 1 Report (New Reviewer)
Comments and Suggestions for Authors
Dear,
I am very grateful to you for the invitation to review the manuscript microorganisms-3515252 by Cirio and coauthors, titled “An in vivo viable recovery microbiological study on permanence in saliva of two different probiotics strain administered through chewing gums”.
The manuscript refers to the scientific field covered by the journal and provides experimental data on an interesting topic. The clarity of the objectives is very good, however deficiencyy id in the research purposes. It is obvious that probiotics have health benefits, and their viability is present in saliva. The research was needed to show their effect on bacteria often present in the oral cavity, at least in an in vitro study. The research is good but has many shortcomings in the idea that has not been fully developed.
Line 133: Please provide manufacturers for chemicals and equipment used in the study in the section Material and methods.
Line 223: Write a brief description of the Flow chart of the study design in the text.
Comments on the Quality of English Language
The quality of English does not limit reading.
Author Response
Dear Reviewer,
Thank you for your advice to improve our article. We respond to your comments below.
The English could be improved to more clearly express the research.
The manuscript was revised by a native English speaker.
The manuscript refers to the scientific field covered by the journal and provides experimental data on an interesting topic. The clarity of the objectives is very good, however deficiency id in the research purposes. It is obvious that probiotics have health benefits, and their viability is present in saliva. The research was needed to show their effect on bacteria often present in the oral cavity, at least in an in vitro study. The research is good but has many shortcomings in the idea that has not been fully developed.
The aim of this study was to evaluate the kinetics of probiotic retention in the oral cavity following administration via chewing gum. In probiotic research, particularly in gastrointestinal applications, studies assessing the presence of probiotic strains in biological matrices are commonly referred to as “persistence” or sometimes “recovery” studies. These studies focus exclusively on the microbiological performance of the probiotic strain, independent of any direct effects on the host.
The present study follows this approach, with the primary goal of quantifying the presence and persistence of probiotic cells in saliva after administration via chewing gum. We are aware that a single administration, is not sufficient to induce lasting changes in the oral microbiota. Previous research has shown that probiotics have only a transient effect on bacterial composition and must be administered continuously over time to achieve a lasting effect. (Chandrasekaran, P.; Weiskirchen, S.; Weiskirchen, R. Effects of Probiotics on Gut Microbiota: An Overview. Int. J. Mol. Sci. 2024, 25, 6022. https://doi.org/10.3390/ijms25116022).
At this stage, the probiotic colonization of oral surfaces was not aimed or assessed. This aspect will be explored. In a subsequent study, in which probiotics will be administered continuously over an extended period.
Line 133: Please provide manufacturers for chemicals and equipment used in the study in the section Material and methods.
We added this information in Material and Methods section.
Line 223: Write a brief description of the Flow chart of the study design in the text.
We added this sentence in the description of Figure 1: At the first appointment, each participant chewed one randomly selected chewing gum from the three gums tested in the study. At the second appointment, they chewed one of the two remaining gums, and at the final appointment, the last remaining gum. This process was repeated for each participant, ensuring that all three gums were tested in a randomized order across the different subjects.
Each chewing gum was chewed two hours after breakfast and the oral hygiene routine. Salivary samples were then collected over a two-hour period and analyzed microbiologically.
Reviewer 2 Report (New Reviewer)
Comments and Suggestions for Authors
Comments to Cirio et al.:
I would rephrase the title to be more concise to show the conclusion of the research. For instance, it is not necessary to write "An in vivo viable recovery microbiological study". The word "permanence" should be replaced with a better word. The intention of the manuscript is that adding probiotics through chewing gums results in the presence of these bacteria in the saliva for a long period of time. Actually, the presence of the bacteria was only studied up to 2 h, so prolonged, persistence are not the proper words in this context.
English editing is required.
There is only one experimental figure.
Abstract:
- Line 25: "dynamics" is not the proper word here. Maybe you intended the presence of the given bacteria in the saliva following their administration.
- The reason for choosing these bacteria should be stated in the abstract.
- It is not clear what is "the area under the curve" in line 33.
- Line 33: It says three probiotics, while only the name of two probiotics is mentioned in the abstract.
- Line: Can you use the word "concentrations" for bacteria? I think "level" is a better word.
- Line 36: What do you mean by "free form" of the bacteria?
- Were the 10 individuals followed up? Any adverse effects, diarrhea?
Introduction:
- Line 48: You need to state which obstetric-gynecological disorder.
- Line 71: correct to: "prevalent"
- Line 72: Please name examples of halitosis-causing bacteria.
- Line 84: A bacterium is not "a compound". Please rephrase the sentence.
- Line 98: Please state the activities of xylitol and fluoride.
- Line 127: Add "in vitro" after "both".
- A hypothesis should be added to the Introduction.
- The "aim" sentence should be rephrased.
Material and Methods:
- "viable recovery microbiological study" is not a good description and should be rephrased.
- Line 128: should "n." be "no."?
- Line 146: It is not clear why "a sample size calculation" could not be performed. Are 10 people really sufficient for statistical analysis?
- Line 165: The specific name of the polyols should be stated.
- The exact composition of the chew gums should be stated.
- The sources of all material should be stated.
- The LGG strain should be mentioned, and the working address of Chr. Hansen stated.
- The number of bacteria in each chew gum should be stated.
- The scientific word of the "gum base" should be stated.
- Line 174: The temperature required for melting the gum base should be stated.
- You need to state at which temperature the bacteria were added.
- An image of the gum should be shown.
- The rolling and scoring system should be illustrated.
- Please state the size of the chew gum.
- The full name and source of the Stomacher should be stated.
- The composition of the "Maximum Recovery Diluent Buffer" should be stated.
- The composition of GYEA and RVB-MRS should be stated.
- Line 191: There is an extra point that should be deleted.
- Line 191: State which "interval" ?
- Line 192: You need to state how you made the anaerobic condition.
- Images of the colonies of the specified bacteria should be shown, and how these colonies were distinguished from the colonies of oral bacteria.
- You need also to describe how you cultivated the bacteria prior to incorporating them into the gum.
- The volunteers got three gums. Two are with bacteria. What was the composition of the third gum? (I see later that it is mentioned in the Result section. This should be stated in the method section).
- Would it not be easier to quantify the specified bacteria by real-time PCR using specific primers on saliva-isolated microbial DNA? Such studies would verify your CFU data.
- How did the probiotics affect the entire oral microbial composition?
- Line 240-241 is superfluous.
- Line 244: spelling mistake. Please correct to: "with".
- The formula used for calculating the area under the curve should be shown.
Results:
- Figure 1 appears thrice in the text (one removed, and two added).
- Lines 261-264 should be shown in the Method section, and provide the answer to one of the questions above.
- Line 265 indicates that the "probiotics" used are not naturally inhabits of the oral cavity. So, there is a need to study the consequences of adding these seemingly "oral foreign" bacteria. It also means that the probiotics administered the week or two weeks previously, were gone and did not thrive in the saliva or in the oral cavity. This has to be tested. What is the viability of the probiotics in saliva?
- Figure 2: You should delete time 0 to start at T1. Else it is misleading with the initial horizontal line.
- Is there a correlation between the 5 kinetic curves in the different individuals? It seems that some individuals eliminate better the bacteria than others: 4 with fast elimination, and 6 with slow elimination.
- It would be better only to present the current figures, without showing the deleted ones, to be sure of what you are presenting.
- I would again not say "concentration" in the y-axis. Concentration is used for a solute dissolved in a solution. Level?
- Graphs of the individual participants should be shown.
- The "total" graph is confusing and superfluous.
- A graph summarizing the AUC should be presented.
- Supplementary Table S2: You can't use the word "concentration" of bacteria. Level?
Discussion:
- You did not study "dynamics", but the fate of the bacteria following their release from the chew gums. The text should accordingly be corrected.
- Line 318: I would not call this persistence, when there is a decline. You are speaking about a short survival of the bacteria in the saliva.
- You need to prove that the delivered bacteria have colonized the plaques, and how this affects tooth caries. The text should be rephrased.
- The "peak" is immediately after chewing the gums, and then there is a decline. Where have the bacteria gone? To the GIT? Or are they killed by the saliva? How can such a short exposure to oral cavity-foreign bacteria affect oral health? Please discuss this issue.
- Line 362: You studied the presence and the kinetics of the elimination of the bacteria, not their persistence.
- "Bacterial recovery" from the chewing gums, but not local oral growth.
- Correct all text according to these comments.
Conclusions:
- Line 397: Prolonged is too general wording – the statistics shows that 6 out of 10 still had some of the specified bacteria in the saliva after 2 h, but no later time points were tested. Thus, please tune down the conclusion.

There is a need for English editing.
Author Response
Dear Reviewer,
Thank you for your advice to improve our article. We respond to your comments below.
The English could be improved to more clearly express the research.
The English language has been revised by a native speaker.
I would rephrase the title to be more concise to show the conclusion of the research. For instance, it is not necessary to write "An in vivo viable recovery microbiological study". The word "permanence" should be replaced with a better word. The intention of the manuscript is that adding probiotics through chewing gums results in the presence of these bacteria in the saliva for a long period of time. Actually, the presence of the bacteria was only studied up to 2 h, so prolonged, persistence are not the proper words in this context
Title was modified as follow: In Vivo Study on the Salivary Kinetics of Two Probiotic Strains Delivered via Chewing Gum
English editing is required.
The English language has been revised by a native speaker.
There is only one experimental figure.
We added one figure in the manuscript and another one in supplementary file.
Abstract:
- Line 25: "dynamics" is not the proper word here. Maybe you intended the presence of the given bacteria in the saliva following their administration.
We replaced the word “dynamics” with “kinetics”.
- The reason for choosing these bacteria should be stated in the abstract.
We added a sentence specifying this in the abstract.
- It is not clear what is "the area under the curve" in line 33.
We have added a sentence to better specify what the area under the curve refers to.
- Line 33: It says three probiotics, while only the name of two probiotics is mentioned in the abstract.
The text has been amended
- Line: Can you use the word "concentrations" for bacteria? I think "level" is a better word.
We replaced the word “concentration” with the word “level” in the manuscript
- Line 36: What do you mean by "free form" of the bacteria?
"Free form" refers to the non-microencapsulated version, and we are specifically referring to Lacticaseibacillus rhamnosus GG, which is administered in one chewing gum in a microencapsulated form and in another chewing gum in a non-microencapsulated (free form).
- Were the 10 individuals followed up? Any adverse effects, diarrhea?
We added a sentence specifying this in the abstract.
Introduction:
- Line 48: You need to state which obstetric-gynecological disorder.
We replaced the word “obstetric-gynecological” with “pregnancy”.
- Line 71: correct to: "prevalent"
We replaced the word “prevalence” with “prevalent”
- Line 72: Please name examples of halitosis-causing bacteria.
We added the name of halitosis-causing bacteria in the text.
- Line 84: A bacterium is not "a compound". Please rephrase the sentence.
The sentence was rephrased.
- Line 98: Please state the activities of xylitol and fluoride.
We added a statement on this item in the text.
- Line 127: Add "in vitro" after "both".
We added “in vitro”
- A hypothesis should be added to the Introduction.
A hypothesis has been added.
- The "aim" sentence should be rephrased.
The aim sentence has been rephrased.
Material and Methods:
- "viable recovery microbiological study" is not a good description and should be rephrased.
We modified with “in vivo microbiological study”
- Line 128: should "n." be "no."?
We replaced “n.” with “no.”
- Line 146: It is not clear why "a sample size calculation" could not be performed. Are 10 people really sufficient for statistical analysis?
A sample size calculation hasn’t been done because there weren’t previous studies on permanence in saliva of probiotic administered via chewing gum.
In the field of probiotic research, particularly for gastrointestinal applications, studies assessing the presence of probiotic strains in biological matrices are commonly referred to as “persistence” or “recovery” studies. These studies focus solely on the microbiological performance of the probiotic strain, independent of any direct effects on the host. Our study aligns with this established approach, as its primary objective was to quantify the presence and persistence of probiotic cells in saliva after administration via chewing gum.
Given this specific focus, the sample size used in our study is fully consistent with existing literature on probiotic recovery studies. Numerous published works in this domain have employed similar sample sizes, as the primary endpoint (bacterial recovery) does not require the large cohorts typically associated with clinical efficacy trials. Furthermore, our crossover design enhances the reliability of our results by allowing within-subject comparisons, thereby reducing inter-individual variability. This approach is well-suited to microbiological recovery studies, where the primary goal is to assess the retention and viability of the probiotic strain in a controlled setting.
- Line 165: The specific name of the polyols should be stated.
We understand the reviewer's request; however, the exact formulation of the chewing gum, including the name of the polyols, must remain confidential until it is patented, which will only happen after the ability of the tested probiotics to colonise dental plaque has also been evaluated (study currently in progress). We have added the indication ‘proprietary blend’ in the manuscript.
- The exact composition of the chew gums should be stated.
We understand the reviewer's request; however, the exact formulation of the chewing gum, including the amount of freeze-dried culture used, must remain confidential until it is patented, as requested from the supplier company that financed the study.
- The sources of all material should be stated.
We added this information in the manuscript.
- The LGG strain should be mentioned, and the working address of Chr. Hansen stated.
We added this information in the manuscript.
- The number of bacteria in each chew gum should be stated.
We stated the number of probiotics contained in each gum at the end of the production process in the results section.
- The scientific word of the "gum base" should be stated.
There is no alternative scientific word for the term ‘gum base; we added the name of the manufacturer in Material and Method section.
- Line 174: The temperature required for melting the gum base should be stated.
We added this information in the manuscript.
- You need to state at which temperature the bacteria were added.
We added this information in the manuscript.
- An image of the gum should be shown.
We added an image of the chewing gum in supplementary materials.
- The rolling and scoring system should be illustrated.
We added this information in the manuscript.
- Please state the size of the chew gum.
We stated the weight of each chewing gum in section “chewing gum producing process”.
- The full name and source of the Stomacher should be stated.
We added this information in the manuscript.
- The composition of the "Maximum Recovery Diluent Buffer" should be stated.
We added this information in the manuscript.
- The composition of GYEA and RVB-MRS should be stated.
We added this information in the manuscript.
- Line 191: There is an extra point that should be deleted.
Thank you, we deleted it.
- Line 191: State which "interval"?
We specified and moved the sentence to the correct place in the text.
- Line 192: You need to state how you made the anaerobic condition.
We added this information in the manuscript.
- Images of the colonies of the specified bacteria should be shown, and how these colonies were distinguished from the colonies of oral bacteria.
We do not possess images of the colonies of the probiotic bacteria tested. The colonies were identified because the media were chromogenic and selective for the specific strains sought. In the manuscript we specified that: ‘CFUs were identified based on morphology, size and colour and finally counted”.
- You need also to describe how you cultivated the bacteria prior to incorporating them into the gum.
We did not cultivate the bacteria; we used commercial strains. We purchased a concentrated probiotic powder, supplied by the suppliers mentioned. The bacteria are cultivated in bioreactors, but this is a proprietary process of which we do not know the details.
- The volunteers got three gums. Two are with bacteria. What was the composition of the third gum? (I see later that it is mentioned in the Result section. This should be stated in the method section).
All three gums tested contained probiotics. Two contained LGG either in free or microencapsulated form and the third chewing gum contained Heyndrickxia coagulans. This was better clarified in M&M.
- Would it not be easier to quantify the specified bacteria by real-time PCR using specific primers on saliva-isolated microbial DNA? Such studies would verify your CFU data.
We chose to use the colony-forming unit (CFU) method on a plate because this study aimed to quantify the number of viable bacterial colonies. Real-time PCR is a technique for the relative quantification of DNA, including non-viable bacteria, which could have led to distorted results.
- How did the probiotics affect the entire oral microbial composition?
The aim of this study was to evaluate the kinetics of probiotics in saliva after their administration via chewing gum. At this stage, we did not want to investigate their effect on the oral microbiota, an effect that will be studied in a later phase of the study.
- Line 240-241 is superfluous.
We deleted this sentence.
- Line 244: spelling mistake. Please correct to: "with".
Thank you, we corrected it.
- The formula used for calculating the area under the curve should be shown.
Statistic analysis was performed with STATA using the following command:
drop if time == 0
sort ID Probiotic time
gen delta_time = time[_n+1] - time if ID == ID[_n+1] & Probiotic == Probiotic[_n+1]
gen mean_log_Concentr = (log_Concentr + log_Concentr[_n+1]) / 2 if ID == ID[_n+1] & Probiotic == Probiotic[_n+1]
gen trapezoid_area = delta_time * mean_log_Concentr
collapse (sum) trapezoid_area, by(ID Probiotic)
mean trapezoid_area, over(Probiotico_num)
pwmean trapezoid_area, over(Probiotico_num) mcompare(tukey) pveffect
Results:
- Figure 1 appears thrice in the text (one removed, and two added).
Thank you, we deleted it.
- Lines 261-264 should be shown in the Method section, and provide the answer to one of the questions above.
We moved this sentence as you suggested
- Line 265 indicates that the "probiotics" used are not naturally inhabits of the oral cavity. So, there is a need to study the consequences of adding these seemingly "oral foreign" bacteria. It also means that the probiotics administered the week or two weeks previously, were gone and did not thrive in the saliva or in the oral cavity. This has to be tested. What is the viability of the probiotics in saliva?
As we said in the previous answers, the purpose of this study was not to investigate the effects of probiotic colonisation on oral surfaces. Previous studies show that a single administration of probiotic bacteria cannot change the composition of the oral bacterial flora except in a very transient manner. Therefore, we believe that a washout period of one week was adequate to ensure baseline-like conditions.
- Figure 2: You should delete time 0 to start at T1. Else it is misleading with the initial horizontal line.
We modified Figure 2
- Is there a correlation between the 5 kinetic curves in the different individuals? It seems that some individuals eliminate better the bacteria than others: 4 with fast elimination, and 6 with slow elimination.
The reviewer is absolutely right. In fact, in the manuscript this aspect was emphasised from the abstract to the conclusion. However, The aim was to compare the kinetics of different probiotic strains, not the different among subjects which analysis would have required a larger sample. Following your suggestion, we have included a sentence in the discussion to emphasise this aspect further
- It would be better only to present the current figures, without showing the deleted ones, to be sure of what you are presenting.
We sent the text with the revisions highlighted according to the Editor's instructions. In the next submission we will submit also a version without the highlighted revisions in order to make it clearer to read.
- I would again not say "concentration" in the y-axis. Concentration is used for a solute dissolved in a solution. Level?
We replaced the word “concentration” with the word “level” in the entire manuscript
- Graphs of the individual participants should be shown.
As we reported above, the purpose of this study is not to make a comparison between kinetics of different individuals, but to make a comparison between kinetics of different probiotic strains. For this reason, we did not consider to include this graph in the manuscript. We added this graph in supplementary file.
The "total" graph is confusing and superfluous.
We have deleted it and replaced it with a graph showing the average curves for each probiotic.
- A graph summarizing the AUC should be presented.
Please, see previous answer
- Supplementary Table S2: You can't use the word "concentration" of bacteria. Level?
We replaced the word “concentration” with the word “level” in the supplementary file
Discussion:
- You did not study "dynamics", but the fate of the bacteria following their release from the chew gums. The text should accordingly be corrected.
We replaced the word “dynamics” with “kinetics” or “presence of probiotic in saliva” in the entire manuscript.
- Line 318: I would not call this persistence, when there is a decline. You are speaking about a short survival of the bacteria in the saliva.
We replaced the word “persistence” with “presence of probiotic in saliva”.
- You need to prove that the delivered bacteria have colonized the plaques, and how this affects tooth caries. The text should be rephrased.
This study is the initial phase of a broader research initiative. In this study, we wanted to test whether chewing gum was able to release probiotics and to what extent. In a second study, chewing gum will be administered for prolonged periods and its effect on plaque colonisation and some clinical variables will be investigated.
The "peak" is immediately after chewing the gums, and then there is a decline. Where have the bacteria gone? To the GIT? Or are they killed by the saliva? How can such a short exposure to oral cavity-foreign bacteria affect oral health? Please discuss this issue.
Certainly, some of the probiotic is readily swallowed. In fact, chewing gum rapidly increases salivary flow and this stimulates the subject to swallow. However, it cannot be excluded that a portion of probiotics may be killed by substances with bactericidal action present in saliva. As previous reported, we are aware that a single administration of probiotics is not enough to stably modify the composition of the oral microflora, but this was not the aim of the present investigation. This aspect is now reported in the discussion.
- Line 362: You studied the presence and the kinetics of the elimination of the bacteria, not their persistence.
The reviewer is right, the word “persistence” was replaced with “presence of probiotic in saliva” or “kinetics.
- "Bacterial recovery" from the chewing gums, but not local oral growth.
The sentence has been changed to avoid a misleading interpretation.
- Correct all text according to these comments.
We thank the reviewer for his helpful suggestions, which we have tried to follow scrupulously, and hope that the changes we have made have sufficiently increased the clarity and quality of the paper.
Conclusions:
- Line 397: Prolonged is too general wording – the statistics shows that 6 out of 10 still had some of the specified bacteria in the saliva after 2 h, but no later time points were tested. Thus, please tune down the conclusion.
We moderated the conclusions
Round 2
Reviewer 1 Report (New Reviewer)
Comments and Suggestions for Authors
The manuscript has been significantly improved and may be acceptable with minor corrections listed below
Line 72 and 88: Please write the full name of bacteria, they are mentioned for the first time.
Line 216: Change the subtitle to “Use of chewing gum”
Author Response
The manuscript has been significantly improved and may be acceptable with minor corrections listed below
Dear Reviewer,
We are pleased you appreciated the edits made to the manuscript. Below the further changes made thanks to your suggestions.
Line 72 and 88: Please write the full name of bacteria, they are mentioned for the first time.
We wrote the full name of bacteria, as you suggested.
Line 216: Change the subtitle to “Use of chewing gum”
We modified the subtitle.
Reviewer 2 Report (New Reviewer)
Comments and Suggestions for Authors
The manuscript has been improved and is suitable for publication.
Minor comments:
- Line 248: Please provide the composition or source of the MDR buffer.
- Please provide all letters and the axes in the figures in black instead of gray color. The letters in the figure should be made larger and readable.
- In the response letter, the authors showed how they calculated the area under the curve. This calculation should be added to the method section.

Author Response
The manuscript has been improved and is suitable for publication.
Dear Reviewer,
We are pleased you appreciated the edits made to the manuscript. Below the further changes made thanks to your suggestions.
Minor comments:
- Line 248: Please provide the composition or source of the MDR buffer.
We added the composition of the MDR.
- Please provide all letters and the axes in the figures in black instead of gray color. The letters in the figure should be made larger and readable.
We modified figures as you suggested.
- In the response letter, the authors showed how they calculated the area under the curve. This calculation should be added to the method section.
We added the calculation in material and method section.
This manuscript is a resubmission of an earlier submission. The following is a list of the peer review reports and author responses from that submission.
Round 1
Reviewer 1 Report
Comments and Suggestions for Authors
The article presented is interesting, well-illustrated and addresses a topic of particular interest, especially at the pharmacological level. It interestingly tackles a new topic, although it is a pilot study with a low number of subjects enrolled, but it opens up useful insights and ideas for the future by capturing an aspect of probiotic strain administration that is as simple as ingenious.
Below are reported some small suggestions for the paper:
- In the introduction, line 93-94, the authors cite only the studies done in vitro and in vivo on the administration of probiotics through sugar-free chewing gum. I suggest a more detailed description, brief but highlighting aspects deemed interesting for the purpose of the paper.
- Is it possible to better specify how microencapsulation of L. ramnosus occurs?
- Line 158: please specify in CFU/g that g means “gum”
- In the conclusion/discussion section: please provide a better explanation of the future intentions to continue this work. Could the administration affect the oral microbiota? Are there any studies on this? Are they planning to analyze the oral microbiota at different times?
Author Response
We want to thank the reviewers for the support provided to improve our manuscripts.
Below the comments of the reviewers in bold and our replies in italics.
The article presented is interesting, well-illustrated and addresses a topic of particular interest, especially at the pharmacological level. It interestingly tackles a new topic, although it is a pilot study with a low number of subjects enrolled, but it opens up useful insights and ideas for the future by capturing an aspect of probiotic strain administration that is as simple as ingenious.
Below are reported some small suggestions for the paper:
- In the introduction, line 93-94, the authors cite only the studies done in vitro and in vivo on the administration of probiotics through sugar-free chewing gum. I suggest a more detailed description, brief but highlighting aspects deemed interesting for the purpose of the paper.
Thank you for your suggestion, a brief description was added to the introduction section.
- Is it possible to better specify how microencapsulation of L. ramnosus occurs?
The type of microencapsulation has been provided in the text
- Line 158: please specify in CFU/g that g means “gum”
Cfu/g means colony-forming units /gram, we specified that in the manuscript
- In the conclusion/discussion section: please provide a better explanation of the future intentions to continue this work. Could the administration affect the oral microbiota? Are there any studies on this? Are they planning to analyze the oral microbiota at different times?
An explanation of future studies has been widened in the conclusions section.
Reviewer 2 Report
Comments and Suggestions for Authors
The introduction in this article sufficiently reflects the problem that the authors want to raise in their study.
However, starting with the materials and methods, I find many points of contention.
The authors do not justify why they chose these particular bacterial cultures.
In the materials and methods the authors do not explain what encapsulated form of Lacticaseibacillus rhamnosus is, I did not find this product on the Internet, there is no country of manufacture. In addition, the authors do not give the formulation of the chewing gum, and also how many mg of lyophilically dried culture they apply according to the formulation.
Further, the study design and representativeness of the experiment raises many questions. From the point of view of statistics, the experiment does not stand up to criticism.
From the point of view of the experiment, the scheme in Figure 3 is not very clear.
Figure 2 has the word kinetics in the title, it is actually dynamics as the authors present the change in abundance.
The article should be statistically more rigorous and needs a major revision in methodological terms!
At the moment I cannot recommend it for publication.
Author Response
We want to thank the reviewers for the support provided to improve our manuscripts.
Below the comments of the reviewers in bold and our replies in italics.
The introduction in this article sufficiently reflects the problem that the authors want to raise in their study.
However, starting with the materials and methods, I find many points of contention.
The authors do not justify why they chose these particular bacterial cultures.
We appreciate the reviewer's valuable comment. The introduction has been revised to provide a clearer introduction to the tested microorganisms, highlighting the key aspects that justify their selection for the trial. In brief, the bacteria were selected according to two criteria: the scientific literature evidence, but also safety of use (QPS list presence) and industrial availability.
We realised that in many strains whose properties are reported in scientific studies, they are not commercially available. With a view to continuing the studies and one day bringing the benefits found to the population, this aspect is also fundamental.
In the materials and methods the authors do not explain what encapsulated form of Lacticaseibacillus rhamnosus is, I did not find this product on the Internet, there is no country of manufacture.
The type of microencapsulation has been specified in accordance with the manufacturer's claims.
In addition, the authors do not give the formulation of the chewing gum, and also how many mg of lyophilically dried culture they apply according to the formulation.
We understand the reviewer's request; however, the exact formulation of the chewing gum, including the amount of freeze-dried culture used, must remain confidential until patented, as per the request of the supplying company that also funded the study. Nevertheless, we have added a list of the main ingredient categories in the manuscript and specified the amount of probiotic present in the chewing gum at the end of the manufacturing process.
Further, the study design and representativeness of the experiment raises many questions. From the point of view of statistics, the experiment does not stand up to criticism.
Our study was designed as a pilot study, with a limited number of participants (n=10), which is justifiable for exploring a new approach. In specific:
- This is a pilot study, whose primary objective is to generate preliminary data for future studies with larger cohorts.
- The selection of participants was based on strict criteria (e.g., number of teeth, salivary flow, oral health status) to minimize uncontrolled variability.
- Pilot studies with 10 participants are common for testing feasibility and obtaining initial data on new probiotic formulations (Sanctuary MR, Kain JN, Chen SY, Kalanetra K, Lemay DG, Rose DR, Yang HT, Tancredi DJ, German JB, Slupsky CM, Ashwood P, Mills DA, Smilowitz JT, Angkustsiri K. Pilot study of probiotic/colostrum supplementation on gut function in children with autism and gastrointestinal symptoms. PLoS One. 2019 Jan 9;14(1):e0210064. doi: 10.1371/journal.pone.0210064. PMID: 30625189; PMCID: PMC6326569. Grusovin MG, Bossini S, Calza S, Cappa V, Garzetti G, Scotti E, Gherlone EF, Mensi M. Clinical efficacy of Lactobacillus reuteri-containing lozenges in the supportive therapy of generalized periodontitis stage III and IV, grade C: 1-year results of a double-blind randomized placebo-controlled pilot study. Clin Oral Investig. 2020 Jun;24(6):2015-2024. doi: 10.1007/s00784-019-03065-x. Epub 2019 Oct 16. PMID: 31620939)
- Finally, the crossover design employed in this study helps mitigate the limitation due to the small sample, while within-subject comparisons further reduce inter-subject variability.
From the point of view of the experiment, the scheme in Figure 3 is not very clear.
There is no Figure 3 in the manuscript, so we believe you may be referring to Figure 1 or Figure S1. Both figures have been modified for improved clarity.
Figure 2 has the word kinetics in the title, it is actually dynamics as the authors present the change in abundance.
Thank you for your suggestion. We have replaced the word "kinetic" with "dynamic" in both the figure and the text.
The article should be statistically more rigorous and needs a major revision in methodological terms!
The statistical analysis has been comprehensively revised. Should the modifications made prove inadequate, we would be grateful if you could kindly provide further details regarding the additional corrections required.
At the moment I cannot recommend it for publication.
We regret the reviewer's harsh comment and can only hope that the revisions will help change his/her opinion.
Reviewer 3 Report
Comments and Suggestions for Authors
In the manuscript "An in vivo pilot study on permanence in saliva of two different probiotics strains administered through chewing gums" the authors present the results of the pilot study. Although I really liked the idea of the research as well as the combination of bacterial preparations and the application of probiotics via chewing gum has potential, I have comments on the design of the study. The main objection would be the small number of adult volunteers, but the authors clearly show and explain this. My objection goes in the direction that 5 application protocols of bacteria in different forms (vegetative, microencapsulated and spore) and application protocol were applied to 10 subjects. It is understandable that in the end it was not commented at all which combination or order of application achieves a better result. Furthermore, how did you distinguish the microencapsulated L. rhamnosus GG bacteria from the vegetative one by doing a tenfold dilution series and counting the grown colonies? How did you get a negative number from one of the respondents?
Author Response
We want to thank the reviewers for the support provided to improve our manuscripts.
Below the comments of the reviewers in bold and our replies in italics.
In the manuscript "An in vivo pilot study on permanence in saliva of two different probiotics strains administered through chewing gums" the authors present the results of the pilot study. Although I really liked the idea of the research as well as the combination of bacterial preparations and the application of probiotics via chewing gum has potential, I have comments on the design of the study.
We appreciate the reviewer's positive feedback on our study. We would like to clarify that no combination of probiotics was administered; each strain was tested individually. It is possible that the original flowchart caused some misinterpretation of the study design. To address this, we have revised the flowchart, which is now included in the paper.
The main objection would be the small number of adult volunteers, but the authors clearly show and explain this. My objection goes in the direction that 5 application protocols of bacteria in different forms (vegetative, microencapsulated and spore) and application protocol were applied to 10 subjects. It is understandable that in the end it was not commented at all which combination or order of application achieves a better result.
Our study was designed as a pilot study, with a limited number of participants (n=10), which is justifiable for exploring a new approach. In specific:
- This is a pilot study, whose primary objective is to generate preliminary data for future studies with larger cohorts.
- The selection of participants was based on strict criteria (e.g., number of teeth, salivary flow, oral health status) to minimize uncontrolled variability.
- Pilot studies with 10 participants are common for testing feasibility and obtaining initial data on new probiotic formulations (Sanctuary MR, Kain JN, Chen SY, Kalanetra K, Lemay DG, Rose DR, Yang HT, Tancredi DJ, German JB, Slupsky CM, Ashwood P, Mills DA, Smilowitz JT, Angkustsiri K. Pilot study of probiotic/colostrum supplementation on gut function in children with autism and gastrointestinal symptoms. PLoS One. 2019 Jan 9;14(1):e0210064. doi: 10.1371/journal.pone.0210064. PMID: 30625189; PMCID: PMC6326569. Grusovin MG, Bossini S, Calza S, Cappa V, Garzetti G, Scotti E, Gherlone EF, Mensi M. Clinical efficacy of Lactobacillus reuteri-containing lozenges in the supportive therapy of generalized periodontitis stage III and IV, grade C: 1-year results of a double-blind randomized placebo-controlled pilot study. Clin Oral Investig. 2020 Jun;24(6):2015-2024. doi: 10.1007/s00784-019-03065-x. Epub 2019 Oct 16. PMID: 31620939)
- Finally, the crossover design employed in this study helps mitigate the limitation due to the small sample, while within-subject comparisons further reduce inter-subject variability.
Furthermore, how did you distinguish the microencapsulated L. rhamnosus GG bacteria from the vegetative one by doing a tenfold dilution series and counting the grown colonies?
Saliva samples collected after the administration of microencapsulated LGG were not subjected to any treatment to disaggregate the microcapsules. Consequently, only the microorganisms that were free from the microcapsule were able to proliferate after plating. We have clarified this point in the text
How did you get a negative number from one of the respondents?
In the Log transformation, we have rounded values of 0 to 1 to avoid negative values. We have specified this fact in materials and methods section.
Round 2
Reviewer 2 Report
Comments and Suggestions for Authors
The authors have greatly improved the article, especially in terms of the validity of these studies.
Author Response
The authors have greatly improved the article, especially in terms of the validity of these studies.
Dear, Reviewer, we are pleased that the reviewer appreciates the revisions made to the paper and thank him/her for his/her valuable feedback.
Reviewer 3 Report
Comments and Suggestions for Authors
The authors have worked hard to improve the quality of the article, but there are still unanswered questions. In the first version of the article, there is a diagram showing that two subjects took three gummies in different orders. How did we put them in the same basket if they did not take the same gummies in the same order? I am very aware that there are pilot studies and that they are publishable with a small number of subjects, but this is not the case here. Do we have controls?
Why do we have a separate graph for L. rhamnosus GG in microencapsulated form? How did you determine the number of LGG in microencapsulated form in the saliva of the subject, even though he received LGG in free form?
The graphs do not indicate the unit in which the results are expressed?
Author Response
We want to thank the reviewers for the support provided to improve our manuscripts.
Below the comments of the reviewers in bold and our replies in italics.
The authors have worked hard to improve the quality of the article, but there are still unanswered questions.
- In the first version of the article, there is a diagram showing that two subjects took three gummies in different orders. How did we put them in the same basket if they did not take the same gummies in the same order?
We modified the diagram in Figure 1 because we realized that it could be misleading. Specifically, each participant at the first appointment extracted one of three chewing gum using a lottery method. At the second appointment, he/she drew one of the two remaining chewing gums, and at the third appointment, the last remaining chewing gum was administered. We have clarified this in the manuscript.
- I am very aware that there are pilot studies and that they are publishable with a small number of subjects, but this is not the case here. Do we have controls?
We appreciate your thoughtful comments and the opportunity to clarify the nature of our study. After careful reflection on your feedback, we recognize that the term "pilot study" may not be the most appropriate descriptor for our work. Instead, our study should be more precisely categorized as a viable recovery microbiological study.
In the field of probiotic research, particularly for gastrointestinal applications, studies assessing the presence of probiotic strains in biological matrices are commonly referred to as “persistence” or “recovery” studies. These studies focus solely on the microbiological performance of the probiotic strain, independent of any direct effects on the host. Our study aligns with this established approach, as its primary objective was to quantify the presence and persistence of probiotic cells in saliva after administration via chewing gum.
Given this specific focus, the sample size used in our study is fully consistent with existing literature on probiotic recovery studies. Numerous published works in this domain have employed similar sample sizes, as the primary endpoint (bacterial recovery) does not require the large cohorts typically associated with clinical efficacy trials. Furthermore, our crossover design enhances the reliability of our results by allowing within-subject comparisons, thereby reducing inter-individual variability. This approach is well-suited to microbiological recovery studies, where the primary goal is to assess the retention and viability of the probiotic strain in a controlled setting.
For your question regarding controls. In microbiological viable recovery studies, the most relevant control is the confirmation that the probiotic strain of interest is absent in the biological matrix before administration. In our study, we ensured this by verifying that none of the participants had detectable levels of the tested probiotic strains in their saliva prior to chewing the gum (T0 time point).
A control group using a probiotic-free chewing gum would not provide additional meaningful information in this context, as our study does not aim to evaluate clinical effects or host responses, but rather the persistence of the administered probiotic in saliva. The presence or absence of probiotics in post-administration samples can be directly attributed to the intervention itself, as confirmed by the absence of these strains in baseline samples.
We hope this clarification adequately addresses your concern. We appreciate your insights and remain open to further discussion.
- Why do we have a separate graph for L. rhamnosus GG in microencapsulated form? How did you determine the number of LGG in microencapsulated form in the saliva of the subject, even though he received LGG in free form?
Thank you for your question. We would like to clarify that in our study, Lacticaseibacillus rhamnosus GG (LGG) was tested in two distinct formulations: a free-form lyophilized preparation and a microencapsulated preparation. These two formulations are industrially distinct, with the microencapsulated version undergoing an additional technological process designed to enhance its stability and survival during storage and delivery.
Given that these are two separate preparations, they were analyzed independently to answer two key research questions:
- Whether L. rhamnosus GG is a suitable probiotic for oral administration via chewing gum.
- Whether microencapsulation improves its persistence in saliva, which is critical for evaluating its potential advantages over the free form.
At any time- point we attempt to distinguish between microencapsulated and free-form LGG within the same subject. In each time point each subject received only one formulation per session, in a randomized crossover design, ensuring that the viability of each preparation was assessed independently.
Regarding your specific concern:
- The separate graph for LGG in microencapsulated form is necessary because this preparation was tested separately from the free form. It was not derived from subjects who received the free-form LGG, but rather from those who received the microencapsulated version.
- At no point did we attempt to distinguish between microencapsulated and free-form LGG within the same subject. Each subject received only one formulation per session, in a randomized crossover design, ensuring that the viability of each preparation was assessed independently.
We hope this clarification resolves any confusion. Thank you for your insightful feedback, which has allowed us to further refine the manuscript’s clarity.
The graphs do not indicate the unit in which the results are expressed?
Thanks for your advice, we added the unit in which the results are expressed in Figure 2